REGISTERED REPORT PROTOCOL

# Improving dog training methods: Efficacy and efficiency of reward and mixed training methods

**Ana Catarina Vieira de Castro** [1,2]*, **Ângelo Araújo**[3], **André Fonseca**[4], **I. Anna S. Olsson**[1,2]

**1** Instituto de Biologia Molecular e Celular, Universidade do Porto, Porto, Portugal, **2** i3S –Instituto de Investigação e Inovação em Saúde, Universidade do Porto, Porto, Portugal, **3** Polícia de Segurança Pública, Lisbon, Portugal, **4** CINAMIL, The Military Academy Research Center of the Portuguese Army, Lisbon, Portugal

\* ana.castro@ibmc.up.pt

## Abstract

Dogs play an important role in our society as companions and work partners, and proper training of these dogs is pivotal. For companion dogs, training helps preventing or managing dog behavioral problems—the most frequently cited reason for relinquishing and euthanasia, and it promotes successful dog-human relationships and thus maximizes benefits humans derive from bonding with dogs. For working dogs, training is crucial for them to successfully accomplish their jobs. Dog training methods range widely from those using predominantly aversive stimuli (aversive methods), to those combining aversive and rewarding stimuli (mixed methods) and those focusing on the use of rewards (reward methods). The use of aversive stimuli in training is highly controversial and several veterinary and animal protection organizations have recommended a ban on pinch collars, e-collars and other techniques that induce fear or pain in dogs, on the grounds that such methods compromise dog welfare. At the same time, training methods based on the use of rewards are claimed to be more humane and equally or more effective than aversive or mixed methods. This important discussion, however, has not always been based in solid scientific evidence. Although there is growing scientific evidence that training with aversive stimuli has a negative impact on dog welfare, the scientific literature on the efficacy and efficiency of the different methodologies is scarce and inconsistent. Hence, the goal of the current study is to investigate the efficacy and efficiency of different dog training methods. To that end, we will apply different dog training methods in a population of working dogs and evaluate the outcome after a period of training. The use of working dogs will allow for a rigorous experimental design and control, with randomization of treatments. Military (n = 10) and police (n = 20) dogs will be pseudo-randomly allocated to two groups. One group will be trained to perform a set of tasks (food refusal, interrupted recall, dumbbell retrieval and placing items in a basket) using reward methods and the other group will be trained for the same tasks using mixed methods. Later, the dogs will perform a standardized test where they will be required to perform the trained behaviors. The reliability of the behaviors and the time taken to learn them will be assessed in order to evaluate the efficacy and efficiency, respectively, of the different training methods. This study will be performed in collaboration with the Portuguese Army

**Data Availability Statement:** All relevant data from this study will be made available upon study completion.

**Funding:** The authors received no specific funding for this work.

**Competing interests:** The authors have declared that no competing interests exist.

and with the Portuguese Public Security Police (PSP) and integrated with their dog training programs.

## 1. Introduction

The methods used to train dogs range broadly with some using rewards and other non-invasive techniques (reward methods), others using mainly aversive stimuli (aversive methods) and still others using a combination of both (mixed methods). Strong claims have been made for the negative effect of the use of aversive stimuli in training on dog welfare and dog-owner bond. However, the scientific evidence for this has been limited as most studies lack objective welfare measures, investigation of the entire range of aversive techniques and companion dog-focused research [1]. Recently, in the first large-scale quasi-experimental study of companion dog training (n = 92), Vieira de Castro et al (2020) [2] found that dogs trained with aversive stimuli displayed more stress behaviors during training, showed higher elevations in cortisol levels after training and, if trained exclusively with aversive methods, were more 'pessimistic' in a cognitive bias task than dogs trained with either reward and mixed methods. These findings strongly suggest that using aversive stimuli in training compromises companion dog welfare both within and outside the training context. In parallel, in a study aimed at assessing the relationship between training methods and dog-owner bond, Vieira de Castro el al (2019) [3] found that a secure attachment tended to be more consistent in dogs trained with reward methods, as revealed by behaviors displayed during a Strange Situation Procedure. These results suggest that the choice of training methods may also affect dog attachment to owner.

In addition to the effects on welfare, efficacy and efficiency are also relevant aspects to consider for the choice of training methods. Although claims have been made that reward and aversive/mixed methods are, at least, equally effective, the existing scientific literature is inconsistent. Some studies examined the efficacy (reliability of trained behaviors) of specific training methods but without directly comparing reward and aversive/mixed methods. Dale et al (2017) [4] found that dogs learned to avoid native birds after training using e-collars, an aversive technique, and that learning was retained for most dogs following one year. On the other hand, Yin et al (2008) [5] demonstrated that dogs could be trained with a remote-controlled food reward dispenser not to bark excessively, jump and crowd around the door when people arrived. Also, three proof-of-concept studies have shown that clicker training (a reward technique) is effective for training dogs for scent detection tasks [6, 7] and service dog tasks [8]. Other studies have directly compared the efficacy of aversive and reward methods in both dogs and horses and these have produced conflicting results. Among these, five studies suggest a higher efficacy of reward methods [9–13], whereas one points in the opposite direction [14] and three show no differences between methods [15–17]. To our knowledge, only one study addressed the efficiency (speed of learning) of different methods and suggests a higher efficiency of reward over aversive methods [18].

Therefore, the aim of the current study is to evaluate the efficacy and efficiency of different dog training methods. This will be investigated in the context of working dogs, as working dogs allow a rigorous experimental design and control, with randomization of treatments. Namely, military and police dogs will be trained using either reward (Group Reward) or mixed methods (Group Mixed, dogs pseudo-randomly allocated to groups) to perform a set of behaviors. The efficiency of training methods will be evaluated by measuring the number of sessions required for the dogs to learn the tasks, and efficacy will be assessed using a standardized test in which dogs will be required to perform the trained behaviors.

Dogs play an important role in our society both as companion and working animals. Owning a dog for companionship has been shown to bring several physical and psychological benefits to humans [19, 20], and working dogs are of invaluable help when, for example, they fulfil tasks for disabled people or help in the detection of drugs or explosives. Dog training plays a pivotal role here. First, by preventing or managing dog behavioral problems—the most frequently cited reason for relinquishing and euthanasia [21], it helps to promote successful dog-human relationships and thus maximize the benefits humans derive from bonding with dogs [22]. Secondly, because it is required for working dogs to successfully accomplish their jobs.

## 2. Material and methods

### 2.1. Ethics statement

The planned study includes an experimental training protocol in which working dogs are trained with either reward or mixed methods. The mixed methods will be based on the training method presently used for training these dogs outside the experimental protocol, thus no dog will be subjected to pain, suffering, distress or lasting harm as a result of being recruited for the study. Shock collars and pinch collars, which can cause physical harm, will not be used.

Dogs and handlers will be video recorded for further analysis of behavior. Individual handlers will be identifiable from the video footage. Material in which individuals can be identified will only be used by the research team for research purposes (i.e., to control for the training techniques and for data analysis).

All handlers will be briefed that the purpose of the study is "to investigate different training methods and measure the behavior of the dog-handler dyad", and sign an informed consent form that they agree to participate in the study and to be video recorded for research purposes. Each handler will be instructed about which tools and techniques are included in the treatment assigned to them, but will not be informed about the overall experimental design.

Applications for approval are submitted to the Committee for Ethics and Responsible Conduct in Research (human subjects research) and from the Animal Welfare and Ethics Body (animal research) of i3S, University of Porto. The study will only start after approval has been obtained.

### 2.2. Subjects

Military (n = 10) and police dogs (n = 20), housed at the facilities of the Military Working Dog Platoon in the Portuguese Paratroopers Regiment (RPara) and Portuguese Public Security Police (PSP) K9 unit, respectively will be allocated to Group Reward (trained with reward methods) and Group Mixed (trained with mixed methods). All dogs have previous mixed methods training experience, a stratified randomization method [23] will be used to assign animals to the two groups. This method allows for balancing in terms of subjects' baseline characteristics (covariates) that may potentially affect the dependent variables under study. In the present study the following covariates will be taken into account: dog sex, age, breed and previous training experience (obedience, odor detection, protection work). This will be done for each institution, meaning that five dogs from RPara and 10 dogs from the PSP K9 unit will be allocated to each group.

As part of their certification process as working dogs, all the animals had to perform and pass the obedience component of a BH test [24]. Despite all dogs being naïve to the specific exercises included in the present study (food refusal, interrupted recall, dumbbell retrieval and placing items in basket–the detailed description of the exercises is presented below), two similar behaviors are trained as part of the training programs of PSP and RPara. Namely, dogs are trained to retrieve a motivator (e.g., a tug or bite pad), although not to the formality and

precision that is going to be required in the 'dumbbell retrieve' exercise, and they are also usually trained to interrupt a send away (i.e., they are trained to run forward to a motivator and interrupt the running when instructed). The 'food refusal' and 'place items in the basket' exercises are not part of the training programs and are thus new or near to completely new for all the animals. Because previous training on similar behaviors may have carryover effects on the training planned for the study, at the time of the beginning of the study, each participating dog's training history will be thoroughly evaluated and, if needed, this will also be included as a covariate in the randomization process.

## 2.3. Training methods

All dogs will be trained through associative learning (classical and operant conditioning) [25, 26], however, the principles used for each group will differ. Whereas all four quadrants of operant conditioning will be allowed for Group Mixed (positive punishment, negative reinforcement, positive reinforcement and negative punishment), only the quadrants of positive reinforcement and negative punishment will be permitted for Group Reward. Regarding classical conditioning, the use of both conditioned reinforcers and punishers will be allowed for Group Mixed, but only conditioned reinforcers will be allowed for Group Reward. Table 1 displays the detailed definitions for all the conditioning procedures and includes some practical examples.

As for training equipment, no pinch nor e-collars will be allowed in the study and choke chains will only be allowed for Group Mixed. Apart from this, the handlers will be free to decide which other equipment to use among leashes, flat collars and harnesses. The use of a clicker will also be optional, as it has been reported not to affect efficiency and efficacy as compared to the use of a verbal marker or food alone [27–29]. In order to ensure that the instructions regarding the training procedures and tools permitted for each group are being followed, checkpoints will be done at the fifth and tenth days of training for each dyad, when the research team will review the video recordings of the training sessions.

Some flexibility for choosing training equipment and procedures will thus be allowed (as opposed to have the handlers following previously defined and detailed training protocols). The reason for this decision is that this study aims to reflect a real-life situation of dog training,

**Table 1. Definition of the conditioning procedures used for training dogs.**

|  | Procedure | Definition |
|---|---|---|
| **Operant conditioning** | **Positive punishment** | Any unpleasant stimulus that is applied to the dog after the exhibition of an undesirable behavior. Examples include applying a leash jerk, yelling at the dog and leaning towards the dog in a threatening way. |
|  | **Negative reinforcement** | Any unpleasant stimulus that is applied to the dog and that is stopped only after the dog exhibits the desired behavior. Examples include releasing leash pressure. |
|  | **Positive reinforcement** | Any pleasant stimulus that is applied to the dog after the exhibition of a desirable behavior. Examples include food treats, playing tug-of-war, verbal praise, and petting the dog. |
|  | **Negative punishment** | Any pleasant stimulus that is removed after the exhibition of an undesirable behavior. Examples include a time-out in a crate. |
| **Classical conditioning** | **Conditioned punisher** | Any (initially) neutral stimulus that, after repeated paring with an unpleasant stimulus, acquires its properties as a punisher. Examples includes a verbal marker 'no' that was paired with a slap. |
|  | **Conditioned reinforcer** | Any (initially) neutral stimulus that, after repeated paring with a pleasant stimulus acquires its properties as a reinforcer. Examples includes a clicker (a device that makes a clicking sound) that was paired with food delivery. |

where different handlers use different approaches (within the same training method–reward or mixed) and, especially, where the individual dog and its natural tendencies and behaviors usually dictate the training pathway.

## 2.4. Data collection

**2.4.1. Training.** Dogs will be trained by their handlers to perform four exercises: 'food refusal, 'interrupted recall', 'dumbbell retrieval' and 'placing items in basket'. The exercises were chosen in order to resemble real working dog tasks, while not interfering with the dogs' daily working duties. Prior to training commencement, the handlers will be instructed on the exercises that they will train the dogs to perform and on the tools and techniques they are allowed to use during training (as explained in detail in the previous section). The handlers will be free to decide whether to train the exercises in parallel or in a sequence, as well as the order in which to train the different exercises. Training sessions will be conducted two days per week, with a gap between training days no longer than three days. Each training session will have a maximum duration of 10 minutes and up to six training sessions can be conducted per day. Within each training day, a break of at least 30 minutes between training sessions will be required.

Training for each exercise will end when the dog reaches the learning criterion (i.e., adequately performs the behavior as determined by the handler) or after a maximum of 45 sessions. Information regarding the number of training sessions, their duration and the behaviors being trained will be annotated by each handler in a notebook (specifically designed for the study) for each training day. In addition, all training sessions will be video recorded.

**2.4.2. Evaluating performance.** The efficiency of the different training methods will be evaluated through the number of training sessions necessary to reach the learning criterion (as determined by the handlers), and the efficacy will be assessed through a standardized test where the dogs will be asked to perform the trained behaviors. The test will be conducted in a fenced enclosure and will include the following exercises:

1. Food refusal: The handler asks the dog to 'stay' (the position in which the dog is left can be either a sit, a down or a stand, according to the handler's choice), walks 10 meters away to a pre-defined/marked location within the field of vision of the dog, and stops with his/her back facing the dog. Afterwards, a helper comes near the dog and throws two pieces of food next to the dog's front legs, one to right side and one to the left side. The handler can use the verbal cue for the dog not to eat before starting the exercise or while the helper is coming within the field.

Cues: 'Sit'/'Down'/'Stand', 'Stay', 'Don't eat'

2. Interrupted recall: The handler asks the dog to 'stay' (the position in which the dog is left can be either a sit, a down or a stand, according to the handler's choice), walks 30 meters away to a pre-defined/marked location, turns to face the dog and recalls the dog, instructing it to stop after roughly half the distance is covered (the position is which the dog stops can be either a sit, a down or a stand, according to the handler's choice).

Cues: 'Sit'/'Down'/'Stand', 'Stay', 'Come', 'Stop'

3. Dumbbell retrieval: With the dog sitting at his/her side, the handler throws the dumbbell to a distance of roughly 10 meters (marked in the floor in order to help) and then instructs the dog to retrieve it. The dog should move towards the dumbbell, pick it up and bring it to the handler, sit in front of him and only release on cue.

Cues: 'Sit', 'Get it', 'Out'

4. Placing items in basket: A basket will be placed in the field and three items will be placed in pre-defined positions in the floor, around the basket, by a helper. The handler will then instruct the dog to place the items in the basket.

Cues: 'Place' (only one cue is allowed for the entire exercise, the handler is not allowed to give additional cues after each item is retrieved)

5. Surprise exercise: The dog will have to perform a dumbbell retrieval with two pieces of food being thrown to the floor next to the dog by a helper before the exercise starts. This exercise will be included in order to test for training generalization.

Cues: 'Sit', 'Don't eat', 'Get it', 'Out'

The starting points for the exercises will be the same for all dogs and will be marked in the floor with a spray. Only verbal cues will be allowed during the test. The aforementioned words/expressions are, however, purely indicative—each handler will be free to choose his or her own cues. During the test, the dogs will not wear any collar or leash and no treats, toys or punishments will be allowed. Handlers will only be allowed to use social reinforcement (praise) between exercises. Additionally, in order to ensure that all dogs perform the test with similar motivation levels, dogs will be fed 12 hours previously to the conduction of the tests and no play or physical exercise will be allowed during this period.

The designs of Exercises 1, 2 and 3 were inspired on the Federation Cynologique International (FCI) dog sports of IGP, Obedience and Mondioring [24, 30, 31]. Exercise 4 is not part of any recognized dog sport, but its core behavior is (retrieve).

The test will be performed twice, the day after the learning criterion is achieved for all behaviors and 6 months later, to assess short- and long-term efficacy. No formal training will be performed between the two evaluations for 'Food refusal' and 'Placing items in basket'. 'Interrupted Recall' and 'Retrieve dumbbell' will be trained once a month for maintenance. This will be done in order to evaluate the impact of maintenance training on long-term efficacy. The tests will be recorded using two video cameras, set in order to cover the entire field.

A pilot study using two dog-handler dyads that will not participate in the main study will be performed in order to test and, if needed, refine the methodology.

## 2.5. Data analysis

Two different approaches will be used to analyze the performance of the dogs in the test. Three international experts on working dog training will be invited to assess dog performance *in situ* on the test days. The experts, who will be blind to the experimental groups and to the goals of the study, will be instructed to use a qualitative scoring system, according to which the dog performance for each exercise should be classified as 'insufficient', 'sufficient' or 'outstanding' (see S1 Annex for full details). Moreover, two researchers blind to the experimental groups and to the goals of the study will analyze the videos of the tests using a quantitative scoring system, following which the dog performance for each exercise will receive a score ranging from 0 to 10 (see S2 Annex for full details). Inter-observer reliability will be calculated for each exercise. The quantitative scoring system was developed based on FCI rules and guidelines for Obedience, Mondioring and IGP trials [24, 30, 31].

The video recordings of the training sessions and the tests will also be used to assess dog welfare through the analysis of stress behaviors as in Vieira de Castro et al (2020). These will also allow for the analysis of handler behavior and other aspects of training such as the frequency, type and timing of the stimuli applied. This will be used to generate a list of all the conditioning procedures actually used by each handler during training.

**2.5.1. Statistical analysis.** Data will be analyzed using a Generalized Linear Mixed Model (GLMM), to account for repeated measures and to investigate the effects of multiple subject variables. Subject ID will be included as the repeated measure. Age (years), sex (M/F), breed and training experience will be included as covariates and Training Method (Mixed vs Reward) and Training Unit (PSP, RPara) as factors. One model will be run for each response

variable: 1) number of training sessions necessary to reach the learning criterion, 2) qualitative score obtained in the test and 3) quantitative score obtained in the test.

## Supporting information

**S1 Annex. Qualitative scoring system for the test for efficacy evaluation.**
(DOCX)

**S2 Annex. Quantitative scoring system for the test for efficacy evaluation.**
(DOCX)

## Author Contributions

**Conceptualization:** Ana Catarina Vieira de Castro, Ângelo Araújo, André Fonseca, I. Anna S. Olsson.

**Data curation:** Ana Catarina Vieira de Castro, I. Anna S. Olsson.

**Funding acquisition:** Ana Catarina Vieira de Castro, Ângelo Araújo, André Fonseca, I. Anna S. Olsson.

**Investigation:** Ana Catarina Vieira de Castro, Ângelo Araújo, André Fonseca, I. Anna S. Olsson.

**Methodology:** Ana Catarina Vieira de Castro, Ângelo Araújo, André Fonseca, I. Anna S. Olsson.

**Project administration:** Ana Catarina Vieira de Castro, Ângelo Araújo, André Fonseca, I. Anna S. Olsson.

**Resources:** Ângelo Araújo, André Fonseca, I. Anna S. Olsson.

**Supervision:** Ana Catarina Vieira de Castro, Ângelo Araújo, I. Anna S. Olsson.

**Validation:** I. Anna S. Olsson.

**Visualization:** Ana Catarina Vieira de Castro.

**Writing – original draft:** Ana Catarina Vieira de Castro.

**Writing – review & editing:** Ângelo Araújo, I. Anna S. Olsson.

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
