## [Decision Letter · Decision Letter 0]

7 Sep 2020

PONE-D-20-23944

Improving dog training methods: Efficacy and efficiency of reward and mixed training methods

PLOS ONE

Dear Dr. Ana Catarina Vieira de Castro,

Thank you for submitting your manuscript to PLOS ONE. After careful consideration, we feel that it has merit but does not fully meet PLOS ONE’s publication criteria as it currently stands. Therefore, we invite you to submit a revised version of the manuscript that addresses the points raised during the review process.

It was reviewed by two experts in the field and they have suggested some modifications be made prior to acceptance.

If you could write a response to reviewers, that will help to expedite revision when you re-submit.

We look forward to receiving your revised manuscript.

I wish you the best of luck with your revisions.

Hope you are keeping safe and well in these difficult times.

Kind regards,

Simon Clegg, PhD

Academic Editor

PLOS ONE

"The author(s) have applied to funding for this work and are currently waiting for decision."

Reviewers' comments:

Reviewer's Responses to Questions

**Comments to the Author**

1. Does the manuscript provide a valid rationale for the proposed study, with clearly identified and justified research questions?

Reviewer #1: Yes

Reviewer #2: Yes

2. Is the protocol technically sound and planned in a manner that will lead to a meaningful outcome and allow testing the stated hypotheses?

Reviewer #1: Yes

Reviewer #2: Partly

3. Is the methodology feasible and described in sufficient detail to allow the work to be replicable?

Reviewer #1: No

Reviewer #2: No

4. Have the authors described where all data underlying the findings will be made available when the study is complete?

Reviewer #1: No

Reviewer #2: Yes

5. Is the manuscript presented in an intelligible fashion and written in standard English?

Reviewer #1: Yes

Reviewer #2: Yes

6. Review Comments to the Author

You may also provide optional suggestions and comments to authors that they might find helpful in planning their study.

Reviewer #1: Dear authors,

Thank you for submitting your protocol to PLOS ONE. Studies of high level of evidence comparing reward-based training methods with aversive-training methods are missing in the literature, your research will hopefully fill an important gap in the field. Overall, your manuscript is well presented, there is a strong rationale for the proposed study and the methodology is feasible and clear. I am looking forward to reading your findings in the future. Please see some of the suggestions I have written below:

Ethics statement: ethical approval of the study protocol is desirable.

Data availability: where will the data of the study be made available? https://journals.plos.org/plosone/s/data-availability (there are some recommendations on this page)

Abstract: you might prefer "randomized" instead of "pseudorandomized" to avoid confusion by readers without a scientific background?

Line 57: what do you mean by "other non-invasive techniques (reward methods)"? It sounds a bit redundant as compared to line 56 ('exclusively rewards') but I may have missed something.

Line 63: in light of the potential importance of this large study, you could write its sample size, e.g. "of companion dog training (n=XXX)"

Line 75: just remove a space here "aversive/mixed methods"

Line 114: in which context are they going to be video recorded? During the whole training process? Analysis of behaviour of both dog and handler?

Line 116: will this material remain confidential or be shared (e.g. supplementary material) upon study publication? To clarify, you could write "will only be used by the research team for research purposes (i.e. data analysis, XXX)".

Lines 123-125: same comment from above about ethical approval

Lines 128-132; 204-210: besides randomization of individuals and control of the variables 'years of training', age and gender, are you going to use any practical measure to reduce group bias? If dogs are likely to differ in performance (e.g. due to age difference, genetics, training experience), a test of baseline performance (e.g. time required to learn a simple new task) could be added.

Lines 159-160: are all dogs naive to these tasks?

Lines 161-162: who will control the number of sessions and duration of sessions? Self-report, diary, video records? How long is each session expected to last? Are you going to standardize the duration of the sessions?

Lines 165-166: at this task, wouldn't be important to tell the handlers to cover "all" the zones of this field as opposed to just letting them walk "randomly"? Is the food going to be spilt on the same areas for all dogs?

Lines 167-168: are they going to use only verbal commands or visual cues are also allowed? Standardization is important.

Lines 169-174: same as above regarding visual and verbal cues.

Line 174: is the food going to be placed on the same area for all the dogs? Any target to help the handlers throw the dumbbell on a similar location?

Line 183: the techniques allowed for each group should be included in the protocol. It lacks details on how exactly the group reward differs from the group mixed - this is a very important piece of information.

Lines 188-190: who is going to score the dogs? Ideally, more than one person should rate the performances and inter-rater reliability should be calculated (especially for the 'general impression' score, as it is more subjective).

Reviewer #2: This protocol covers an interesting and important applied topic: the efficiency and effectiveness of different dog training approaches. However, many methodological details are missing, making it impossible to assess the soundness of the proposed methods.

There are several different reward-based training methods and aversive training methods and, among a given category, they differ in their effectiveness. For example, research has shown that, among reward training methods, diverse methods differ in their efficiency (Fugazza and Miklósi 2014) and effectiveness (Fugazza and Miklósi 2015). I am not aware of studies comparing different aversive methods, but it is logical to assume that, for example, diverse aversive stimuli may differ in the intensity of their effects, at least, and potentially in other aspects.

It is therefore crucial that the authors carefully describe the methods that will be used for training, rather than only classifying them as reward-based or aversive.

A detailed description of the protocol applied in the training sessions with the two methods would help enormously in this sense.

The authors propose that the efficiency of the training methods will be assessed by measuring the number of sessions needed to reach a criterion that is determined by the trainers. However, it is fundamental to know what would be the length (N. of trials? Time?) of a training session. Training sessions of different durations have been shown to produce different outcomes (Demant et al. 2011).

The duration or number of trials of the sessions should be somehow standardized.

It is likely that both the dogs and the handlers of this study will have extensive experience with mixed methods, but little or no experience with reward methods. I think that this may affect the results. How do the authors plan to take it into account?

Since the evaluation of the dogs’ performance in the test is somewhat subjective, I warmly recommend the observer that will score the dogs’ performance to be blind with regard to the treatment received by the dog, to avoid a biased judgment.

References:

Demant H., Ladewig J., Balsby T.J.S., Dabelsteen J. (2011) The effect of frequency and duration of training sessions on acquisition and long-term memory in dogs. Applied Animal Behaviour Science, 133, 228-234.

Fugazza C. and Miklósi A. (2015) Social learning in dog training: the effectiveness of the Do as I do method compared to shaping/clicker training. Applied Animal Behaviour Science, 171, 146-151.

Fugazza, C. and Miklósi Á. (2014) Should old dog trainers learn new tricks? The efficiency of the Do as I do method and shaping / clicker training method to train dogs. Applied Animal Behaviour Science, 153, 53-61.

7. PLOS authors have the option to publish the peer review history of their article (what does this mean?). If published, this will include your full peer review and any attached files.

Reviewer #1: No

---

## [Author Response · Author response to Decision Letter 0]

8 Nov 2020

RESPONSE TO REVIEWERS

We appreciate all the constructive criticism provided by the two anonymous reviewers. In our opinion, the manuscript as it currently stands has improved substantially. In what follows, we present detailed responses to all the comments.

Editor

Some changes were made and we believe the new version of the manuscript fully complies with PLOS ONE’s requirements.

"The author(s) have applied to funding for this work and are currently waiting for decision."

a. Please clarify the sources of funding (financial or material support) for your study. List the grants or organizations that supported your study, including funding received from your institution.

d. If you did not receive any funding for this study, please state: “The authors received no specific funding for this work.”

Please refer to Cover Letter. We have currently no funding approved for the study and hence the statement for financial disclosure at this time should be “The authors received no specific funding for this work.”

We plan on submitting data as supporting information.

Done. Thank you for pointing that out!

Reviewer #1: 

1. Ethics statement: ethical approval of the study protocol is desirable.

We have now submitted our protocol to both the Committee for Ethics and Responsible Conduct in Research (human subjects research) and from the Animal Welfare and Ethics Body (animal research) of i3S, University of Porto.

2. Data availability: where will the data of the study be made available? https://journals.plos.org/plosone/s/data-availability (there are some recommendations on this page)

We plan on submitting data as supporting information.

3. Abstract: you might prefer "randomized" instead of "pseudorandomized" to avoid confusion by readers without a scientific background?

We have now realized that in the first version of the manuscript we did not explain what we meant by pseudo-randomization and, of course, that could be confusing for the readers (especially for those with no scientific background). In the current version of the manuscript, in the section “Subjects”, the following information can be found: 

“A stratified randomization method [23] will be used to assign animals to the two groups. This method allows for balancing in terms of subjects’ baseline characteristics (covariates) that may potentially affect the dependent variables under study. In the present study the following covariates will be taken into account: dog sex, age, breed and previous training experience (obedience, odor detection, protection work).”

In the meantime, we will stick to the use of the word “pseudo-randomly” in the Abstract. In our opinion, using “randomly” can be deceptive, leading the readers to think that we will perform a true random allocation of dogs to the two groups. 

4. Line 57: what do you mean by "other non-invasive techniques (reward methods)"? It sounds a bit redundant as compared to line 56 ('exclusively rewards') but I may have missed something.

With "other non-invasive techniques we meant extinction and negative punishment, other operant conditioning techniques (besides positive reinforcement) that are used within the scope of reward-based methods in dog training. However, we recognize that the way it was phrased could be confusing and we have removed the word “exclusively”. Now it reads: 

“The methods used to train dogs range broadly with some using rewards and other non invasive techniques (reward methods), others using mainly aversive stimuli (aversive methods) and still other using a combination of both (mixed methods).”

5. Line 63: in light of the potential importance of this large study, you could write its sample size, e.g. "of companion dog training (n=XXX)"

We followed the reviewer’s suggestion and changed the text to:

“Recently, in the first large-scale quasi-experimental study of companion dog training (n=92), Vieira de Castro et al (accepted for publication) [2] found that dogs trained with aversive stimuli displayed more stress behaviors during training…”

6. Line 75: just remove a space here "aversive/mixed methods"

Done. Thank you!

7. Line 114: in which context are they going to be video recorded? During the whole training process? Analysis of behaviour of both dog and handler?

We now acknowledge that, in the first version of the protocol, this information was not made clear. The information can now be found in the following places throughout the protocol:

Line 113 Dogs and handlers will be video recorded for further analysis of behavior. Individual handlers will be identifiable from the video footage. Material in which individuals can be identified will only be used by the research team for research purposes (i.e., to control for the training techniques and for data analysis). 

Line 197 all training sessions will be video recorded.

Line 247 The tests will be recorded

Line 258 two researchers blind to the experimental groups and to the goals of the study will analyze the videos of the tests using a quantitative scoring system

Line 263 The video recordings of the training sessions and the tests will also be used to assess dog welfare through the analysis of stress behaviors as in Vieira de Castro et al (accepted for publication). These will also allow for the analysis of handler behavior and other aspects of training such as the frequency, type and timing of the stimuli applied.

8. Line 116: will this material remain confidential or be shared (e.g. supplementary material) upon study publication? To clarify, you could write "will only be used by the research team for research purposes (i.e. data analysis, XXX)".

We will share untreated quantitative data but not the videos. In order to make this information clear, the text now reads: “Material in which individuals can be identified will only be used by the research team for research purposes (i.e., to control for the training techniques and for data analysis”.

8. Lines 123-125: same comment from above about ethical approval

See comment above.

9. Lines 128-132; 204-210: besides randomization of individuals and control of the variables 'years of training', age and gender, are you going to use any practical measure to reduce group bias? If dogs are likely to differ in performance (e.g. due to age difference, genetics, training experience), a test of baseline performance (e.g. time required to learn a simple new task) could be added.

Because, to our knowledge, there is no validated test for evaluating baseline performance, we will not use a test of this sort in order to allocate dogs to groups. However, as can now read in line 150 “as part of their certification process as working dogs, all the animals had to perform and pass the obedience component of a BH test [24]”. This, is our view, already ensures that all the animals have some equivalence in their baseline performance.

10. Lines 159-160: are all dogs naive to these tasks?

This is crucial information and it was missing in the previous version of the manuscript. We have added the following paragraph: “Despite all dogs being naïve to the specific exercises included in the present study (food refusal, interrupted recall, dumbbell retrieval and placing items in basket – the detailed description of the exercises is presented below), two similar behaviors are trained as part of the training programs of PSP and RPara. Namely, dogs are trained to retrieve a motivator (e.g., a tug or bite pad), although not to the formality and precision that is going to be required in the ‘dumbbell retrieve’ exercise, and they are also usually trained to interrupt a send away (i.e., they are trained to run forward to a motivator and interrupt the running when instructed). The ‘food refusal’ and ‘place items in the basket’ exercises are not part of the training programs and are thus new or near to completely new for all the animals. Because previous training on similar behaviors may have carryover effects on the training planned for the study, at the time of the beginning of the study, each participating dog’s training history will be thoroughly evaluated and, if needed, this will also be included as a covariate in the randomization process.”

11. Lines 161-162: who will control the number of sessions and duration of sessions? Self-report, diary, video records? How long is each session expected to last? Are you going to standardize the duration of the sessions?

We have added the following information to the text:

Line 190: “Training sessions will be conducted two days per week, with a gap between training days no longer than three days. Each training session will have a maximum duration of 10 minutes and up to six training sessions can be conducted per day. Within each training day, a break of at least 30 minutes between training sessions will be required.”

Line 196: “Information regarding the number of training sessions, their duration and the behaviors being trained will be annotated by each handler in a notebook (specifically designed for the study) for each training day. In addition, all training sessions will be video recorded.”

12. Lines 165-166: at this task, wouldn't be important to tell the handlers to cover "all" the zones of this field as opposed to just letting them walk "randomly"? 

Is the food going to be spilt on the same areas for all dogs?

We decided to change this exercise to the food refusal exercise of the dog sport of Mondioring. This way, we have both a more standardized exercise and a stronger basis for the scoring.

13. Lines 167-168: are they going to use only verbal commands or visual cues are also allowed? Standardization is important.

Only verbal cues will be allowed. This is now clearly stated in line 234: “Only verbal cues will be allowed during the test.”

14. Lines 169-174: same as above regarding visual and verbal cues.

See response to previous comment.

15. Line 174: is the food going to be placed on the same area for all the dogs?

See response to comment 12 above.

16. Any target to help the handlers throw the dumbbell on a similar location?

Now line 219 reads: “With the dog sitting at his/her side, the handler throws the dumbbell to a distance of roughly 10 meters (marked in the floor in order to help) and then instructs the dog to retrieve it”.

18. Line 183: the techniques allowed for each group should be included in the protocol. It lacks details on how exactly the group reward differs from the group mixed - this is a very important piece of information.

We added a section entitled “Training methods” where this information is now presented. Thank you for pointing this out, this is definitely crucial information.

19. Lines 188-190: who is going to score the dogs? Ideally, more than one person should rate the performances and inter-rater reliability should be calculated (especially for the 'general impression' score, as it is more subjective).

This information can now be found in section “Data analysis”:

“Two different approaches will be used to analyze the performance of the dogs in the test. Three international experts on working dog training will be invited to assess dog performance in situ on the test days. The experts, who will be blind to the experimental groups and to the goals of the study, will be instructed to use a qualitative scoring system, according to which the dog performance for each exercise should be classified as ‘insufficient’, ‘sufficient’ or ‘outstanding’ (see Annex 1 for full details). Moreover, two researchers blind to the experimental groups and to the goals of the study will analyze the videos of the tests using a quantitative scoring system, following which the dog performance for each exercise will receive a score ranging from 0 to 10 (see Annex 2 for full details). Inter-observer reliability will be calculated for each exercise. The quantitative scoring system was developed based on FCI rules and guidelines for Obedience, Mondioring and IGP trials [24, 30, 31].”

Reviewer #2: 

This protocol covers an interesting and important applied topic: the efficiency and effectiveness of different dog training approaches. However, many methodological details are missing, making it impossible to assess the soundness of the proposed methods.

1. There are several different reward-based training methods and aversive training methods and, among a given category, they differ in their effectiveness. For example, research has shown that, among reward training methods, diverse methods differ in their efficiency (Fugazza and Miklósi 2014) and effectiveness (Fugazza and Miklósi 2015). I am not aware of studies comparing different aversive methods, but it is logical to assume that, for example, diverse aversive stimuli may differ in the intensity of their effects, at least, and potentially in other aspects. It is therefore crucial that the authors carefully describe the methods that will be used for training, rather than only classifying them as reward-based or aversive. A detailed description of the protocol applied in the training sessions with the two methods would help enormously in this sense.

Please refer to response to comment #18 of Reviewer #1. We acknowledge that information on training methods was actually missing in the previous version of the manuscript and we have now added a section entitled “Training methods”, where we detail which procedures and tools can be used for each group. We will not use standardized protocols for training and the reasons for this choice are now underpinned in lines 175-180: “Some flexibility for choosing training equipment and procedures will thus be allowed (as opposed to have the handlers following previously defined and detailed training protocols). The reason for this decision is that this study aims to reflect a real-life situation of dog training, where different handlers use different approaches (within the same training method – reward or mixed) and, especially, where the individual dog and its natural tendencies and behaviors usually dictate the training pathway”. However, we believe that the information we have included in the “Training methods” section addresses your concerns. 

2. The authors propose that the efficiency of the training methods will be assessed by measuring the number of sessions needed to reach a criterion that is determined by the trainers. However, it is fundamental to know what would be the length (N. of trials? Time?) of a training session. Training sessions of different durations have been shown to produce different outcomes (Demant et al. 2011). The duration or number of trials of the sessions should be somehow standardized.

Please refer to response to comment #11 of Reviewer #1.

3. It is likely that both the dogs and the handlers of this study will have extensive experience with mixed methods, but little or no experience with reward methods. I think that this may affect the results. How do the authors plan to take it into account?

Reward methods have been used consistently in both institutions (PSP and RPara) for more or less 10 years. We do not foresee any issue regarding the experience of handlers with both methods.

4. Since the evaluation of the dogs’ performance in the test is somewhat subjective, I warmly recommend the observer that will score the dogs’ performance to be blind with regard to the treatment received by the dog, to avoid a biased judgment.

Please refer to response to comment #19 of Reviewer #1.

---

## [Decision Letter · Decision Letter 1]

3 Dec 2020

PONE-D-20-23944R1

Improving dog training methods: Efficacy and efficiency of reward and mixed training methods

PLOS ONE

Dear Dr. Ana Catarina Vieira de Castro,

Thank you for submitting your manuscript to PLOS ONE. After careful consideration, we feel that it has merit but does not fully meet PLOS ONE’s publication criteria as it currently stands. Therefore, we invite you to submit a revised version of the manuscript that addresses the points raised during the review process.

Many thanks for submitting your manuscript to PLOS One

Your manuscript was reviewed by two experts in the field, and they have recommended some minor modifications be made prior to acceptance

I therefore invite you to make these changes and resubmit. If you could write a response to reviewers, that will greatly aid revision upon re-submission

I wish you the best of luck with your revisions

Hope you are keeping safe and well in these difficult times

Thanks

Simon

We look forward to receiving your revised manuscript.

Kind regards,

Simon Clegg, PhD

Academic Editor

PLOS ONE

Reviewers' comments:

Reviewer's Responses to Questions

**Comments to the Author**

1. Does the manuscript provide a valid rationale for the proposed study, with clearly identified and justified research questions?

Reviewer #1: Yes

Reviewer #2: Yes

2. Is the protocol technically sound and planned in a manner that will lead to a meaningful outcome and allow testing the stated hypotheses?

Reviewer #1: Yes

Reviewer #2: Yes

3. Is the methodology feasible and described in sufficient detail to allow the work to be replicable?

Reviewer #1: Yes

Reviewer #2: Yes

4. Have the authors described where all data underlying the findings will be made available when the study is complete?

Reviewer #1: Yes

Reviewer #2: Yes

5. Is the manuscript presented in an intelligible fashion and written in standard English?

Reviewer #1: Yes

Reviewer #2: Yes

6. Review Comments to the Author

You may also provide optional suggestions and comments to authors that they might find helpful in planning their study.

Reviewer #1: Dear authors,

Thank you for considering my suggestions and making several changes to the manuscript. The quality of your work improved substantially, particularly the methodology, which is much clearer and detailed. I am happy to recommend your report protocol for publication and I wish you all the best conducting the study.

Best wishes.

Reviewer #2: The authors have now provided an improved version of the manuscript and I believe that this is now publishable, after minor revision.

Since flexibility is allowed for the trainers to choose which actual rewards and punishments to use, I strongly recommend that in the data collection the authors include a list of the rewards and punishments actually used by the trainers, because these may play an important role in determining the outcome of the training.

Apart from this, which I believe is very important, I only have a few minor suggestions:

Line 62: a parenthesis is missing from the reference.

Lines 151-152: This sentence describing the subjects' previous experience may better fit above (e.g. in line 138). Then you can continue with the more detailed description of similar behaviours previously learnt by the dogs.

Lines 256-257: What will be the criterions for scoring the performance? E.g., speed of execution? Latency to execute? Detailed criterions may help reducing the subjectivity level of this judgment.

7. PLOS authors have the option to publish the peer review history of their article (what does this mean?). If published, this will include your full peer review and any attached files.

Reviewer #2: No

---

## [Author Response · Author response to Decision Letter 1]

3 Feb 2021

Once again, we would like to thank the reviewers for the constructive criticism. The current version of the manuscript addresses the last round of comments by Reviewer #2. We detail them in what follows.

“Since flexibility is allowed for the trainers to choose which actual rewards and punishments to use, I strongly recommend that in the data collection the authors include a list of the rewards and punishments actually used by the trainers, because these may play an important role in determining the outcome of the training.”

We agree with the reviewer. We thought this was made clear in the previous version of the manuscript when we wrote, in lines 265-268, “The video recordings of the training sessions (…) will also allow for the analysis of handler behavior and other aspects of training such as the frequency, type and timing of the stimuli applied”. However, in order to leave no doubt about this idea, we added the current sentence in line 297: “This will be used to generate a list of all the conditioning procedures actually used by each handler during training”.

“Line 62: a parenthesis is missing from the reference.”

Thank you for noticing that! It is now corrected. 

“Lines 151-152: This sentence describing the subjects' previous experience may better fit above (e.g. in line 138). Then you can continue with the more detailed description of similar behaviours previously learnt by the dogs.”

We followed the reviewer’s suggestion and moved this sentence to the beginning of the paragraph (Line 139).

“Lines 256-257: What will be the criterions for scoring the performance? E.g., speed of execution? Latency to execute? Detailed criterions may help reducing the subjectivity level of this judgment.”

We understand the reviewer’s concern here. In a first draft of our qualitative scoring system, we actually had more criteria detailed for each exercise. However, after carefully and thoroughly discussing it with the members of the author team who work very closely and in practice with this type of scoring systems (Ângelo Aráujo and André Fonseca), the end result was the system we proposed in the previous version of the manuscript (Annex S1). Given that the international experts on working dog training that will evaluate dog performance for our study are also familiar with (and actually implement in practice) this type of system, we feel this is actually the best approach for our research.

---

## [Editor Report · Decision Letter 2]

5 Feb 2021

Improving dog training methods: Efficacy and efficiency of reward and mixed training methods

PONE-D-20-23944R2

Dear Dr. Ana Catarina Vieira de Castro,

We’re pleased to inform you that your manuscript has been judged scientifically suitable for publication and will be formally accepted for publication once it meets all outstanding technical requirements.

Kind regards,

Simon Clegg, PhD

Academic Editor

PLOS ONE

Additional Editor Comments:

Many thanks for resubmitting your manuscript to PLOS One

As you have addressed all the comments and the manuscript reads well, I have recommended it for publication

You should hear from the Editorial Office shortly.

It was a pleasure working with you and I wish you the best of luck for your future research

Hope you are keeping safe and well in these difficult times

Thanks

Simon

---

## [Editor Report · Acceptance letter]

10 Feb 2021

PONE-D-20-23944R2 

Improving dog training methods: Efficacy and efficiency of reward and mixed training methods 

Dear Dr. Vieira de Castro:

I'm pleased to inform you that your manuscript has been deemed suitable for publication in PLOS ONE. Congratulations! Your manuscript is now with our production department. 

Kind regards, 

on behalf of

Dr. Simon Clegg 

Academic Editor

PLOS ONE